# An In Vitro Model to Investigate the Potential of Solid Dispersions to Form Pharmacobezoars

**DOI:** 10.3390/pharmaceutics14122608

**Published:** 2022-11-26

**Authors:** Hannes Gierke, Kerstin Schaefer, Lukas Gerlich, Ann-Cathrin Willmann, Verena Bialetzki, Georg Boeck, Teresa Pfrommer, Thomas Nolte, Werner Weitschies

**Affiliations:** 1Department of Biopharmaceutics and Pharmaceutical Technology, Center of Drug Absorption and Transport, University of Greifswald, 17489 Greifswald, Germany; 2Boehringer Ingelheim Pharma GmbH & Co. KG, 88400 Biberach, Germany; 3Boehringer Ingelheim Pharma GmbH & Co. KG, 1120 Vienna, Austria

**Keywords:** pharmacobezoars, in vitro model, spray dried amorphous solid dispersions, preclinical testing, rodent stomach

## Abstract

The formation of pharmacobezoars from suspensions of spray-dried amorphous solid dispersions (SD-ASDs) of new chemical entities (NCEs) and hydroxypropyl methylcellulose acetate succinate (HPMC-AS) represents a non-compound related adverse effect in preclinical oral toxicity studies in rodents. Whereas the contribution of the insolubility of the carrier polymer to this process taking place in the acidic environment of the rodent stomach is conclusive, unawareness of the extent of in vivo pharmacobezoar formation is adverse. In order to evaluate the risk of pharmacobezoar formation before in vivo administration, we subsequently introduce an in vitro model to assess the agglomeration potential of solid dispersions. To verify that the pharmacobezoar formation potential can be assessed based on the observed agglomeration potential, we conducted a sequence of experiments with two HPMC-AS-based SD-ASD formulations. In vitro, we found their different in vivo pharmacobezoar formation potential reflected by a significantly increased agglomerated mass of formulation 1 per day compared to formulation 2. In order to find an approach to reduce the agglomeration potential of solid dispersion from suspensions, we further applied the model to investigate the impact of the viscosity of the vehicle used to prepare suspensions on agglomerate formation.

## 1. Introduction

Appropriate bioavailability is a prerequisite for adequate preclinical testing of new chemical entity (NCE) drug candidate’s safety and efficacy [1]. If sufficient plasma levels following oral administration of crystalline NCE to laboratory animals in preclinical testing cannot be achieved, amorphization is a potent method to increase solubility and dissolution rate. However, preventing recrystallization from the thermodynamically unfavorable amorphous state usually requires additives for the preparation of stable amorphous solid dispersions (ASD). From a broad variety of established carrier polymers for the formation of ASDs for preclinical testing, hydroxypropyl methylcellulose acetate succinate (HPMC-AS) is due to its exceptional potential of maintaining supersaturated concentrations of NCE frequently the polymer of choice [2] that is also well characterized regarding effects on physiologic parameters of test species [3,4,5]. Moreover, the insolubility of the polymer below pH 5.5 can be utilized to prepare suspensions of HPMC-AS-based ASD in an acidic vehicle. These suspensions are valuable for preclinical testing as, contrary to supersaturated solutions of an ASD comprising of an NCE and a pH-independent soluble carrier polymer, their recrystallization tendency is generally smaller due to the solid state of suspended particles prior administration. However, it might be especially the pH-dependent solubility HPMC-AS, preventing ASD particles from dissolution before reaching the small intestine, that led to the previously reported formation of solid pharmacobezoars consisting of agglomerated spray-dried ASD (SD-ASD) particles in the stomach of rodents used in repeated dose oral toxicological studies and studies on embryofetal development [6]. Aside from a few cases in mice, rats were particularly affected by pharmacobezoar formation following repeated oral dosing of suspensions of HPMC-AS-based ASDs. At necropsy, spherical to irregularly shaped pharmacobezoars with diameters even greater than 1 cm were predominantly found in the stomach. Considering the total volume of the rat stomach, ranging between 3.4 and 6.6 mL [7], they took up a significant share of the intragastric volume of the nonglandular or glandular stomach but induced, aside from histopathological findings in the stomach, no relevant symptoms. However, acute symptoms were observed when pharmacobezoars emptied from the stomach induced an obstructive ileus, which required unscheduled euthanasia of three female rats in a 13-week oral toxicity study. A reason for the predisposition of mice and rats to this finding might be the specific anatomy of the rodent stomach (Figure 1), that differs significantly from that of other test species in which no pharmacobezoars were observed.

The lumen of the rodent stomach is separated into two parts, divided by a visually prominent line extending from the esophagus to the greater curvature known as the limiting ridge. It separates the nonglandular part of the stomach from the glandular. The nonglandular section, representing the fundus region of the rat stomach, is grossly characterized by a translucent, elastic wall with a thin layer of smooth muscle [8,9,10]. Following food or fluid uptake, the nonglandular stomach accommodates volume loads without large increases in intragastric pressure [11]. Its volume accounts under continuous feeding conditions for 3/5 of the total stomach volume [10]. Rather than peristaltic movements, a constant tonus pushes the gastric content forward over the limiting ridge into the glandular stomach, where it is mixed by circular muscle contractions [12]. Even though acid secretion is restricted to the mucosa of the glandular section, the intragastric pH in the nonglandular section is just slightly higher. With reported pH values from 3.1 in the glandular section up to 5.1 in the nonglandular stomach, intragastric pH is overall below the pKa of the HPMC-AS L-grade used to prepare SD-ASD [9,13,14,15], thus preventing the carrier polymer of orally administered formulations from dissolution in the rodent stomach. Especially as the occurrence of pharmacobezoars was limited to few preclinical studies whereas no pharmacobezoars were found during necropsy in the majority of studies conducted with HPMC-AS-containing SD-ASDs [6], a project-specific in vitro screening method for the potential of ASDs to form pharmacobezoars during preclinical formulation development became highly recommended to appropriately take care for animal welfare. 

Given the anatomical and physiological heterogeneity within the rat stomach, biorelevant in vitro simulation of the variable, complex dynamic environment is challenging. A series of sophisticated dynamic rat stomach devices has been developed previously to investigate digestion processes in the rat stomach in vitro [16,17,18]. The latest version of this series, the Dynamic in vitro Rat Stomach III, represents a reproduction of a large rat stomach created from silicone rubber in which in vivo peristaltic movements are simulated by an inclined plate compressing the non-glandular part of the silicone mantle and two eccentric wheels compressing the glandular part in between when rotating [18]. Regardless of the differences of peristaltic movements induced in Dynamic in vitro Rat Stomach III to in vivo motility generated by circular contraction waves the glandular portion of the stomach, motions transferred by the rigid inclined plate and eccentric wheels are constant in their spatial extent and cannot adapt to solid intragastric particles present in the lumen, such as agglomerates. 

In this work, we present an in vitro model capable of individual testing of the in vitro agglomeration potential of ASD formulations. First, mechanical function and experimental procedures are introduced in detail, rationales of model development are described and discussed subsequently. 

To investigate the extent of correlation between in vitro agglomeration potential and the pharmacobezoar formation potential in vivo, we applied the in vitro model to comparatively assess the agglomeration potential of two HPMC-AS-based SD-ASDs that revealed a different potential to form pharmacobezoars in preclinical testing. Commonly used to improve stability of suspensions, this sequence of experiments also included testing of the influence of enhanced viscosity of the vehicle used to prepare suspensions in order to find an approach to reduce the agglomeration potential of both formulations.

## 2. Materials and Methods

### 2.1. In Vitro Model

#### 2.1.1. Mechanical Model Description

The central compartments of the in vitro model are the cavities of the 18 syringe cylinders (10 mL Braun Omnifix^®^, B.Braun Melsungen AG, Melsungen, Germany) marked yellow in Figure 2, where the agglomeration of ASD-suspensions proceeds individually in a fluid environment. This compartment is varied in volume by defined movement of the plungers. The extent and velocity of movements can be adjusted by a modified, programmable Landgraf LA 160 syringe pump (Figure 2—red box below the construction), controlled by Syringe Pump Pro software. To introduce the mechanical function of the in vitro model, the following explanations are first restricted to one of the 18 syringes in the model and associated model compounds shown schematically in Figure 3b. One syringe and associated model compounds together represent one unit to test the agglomeration potential of one suspension. Simultaneously, but independent from each other, 18 units can be used. As we usually conducted experiments by testing in triplicate, up to 6 suspensions containing, e.g., different ASDs can be tested parallelly. 

The model implements alternating processes of medium uptake and emptying, simulating intragastric volume changes by variation of the volume of the cavity of the syringe cylinder [13], and from the reservoirs [14]. The reservoirs contain 50 mL medium (deionized water pH 3), which is replaced every day.

For medium uptake, the syringe plunger [11] has to be drawn backwards in the syringe cylinder. Therefore, electro motor of the syringe pump transmits a backward movement via a threaded spindle [4] to the complex of the movement transition bar [1] and retainer bar [9], that both have a fixed distance to each other and induce the movements of the syringe plunger. As the plunger endcaps [7] are adjacent to the retainer bar, the plungers have to directly follow the backward movement and accordingly draw up medium from the reservoirs into the cavities of the syringes.

Emptying processes from the syringe require forward movement of the syringe plunger. To isolate individual syringes in the model, we developed a construction where the forward movement is individually transmitted to every syringe plunger instead of attaching all plungers rigidly to one bar. A key point of this construction (Figure 3b—cross section) is the variable rest of the syringe plunger in the complex of movement transmission bar, springs [3], and the retainer bar [9] with recesses for the syringe plungers [8]. The distance between movement transmission bar and retainer bar is fixed (44 mm), so that both have to move in parallel. The retainer bar can be opened to position the syringe plungers in the respective lower half of the recesses in the bar. Once the top of the retainer bar with the upper half of the recesses is screwed back into position, the cross-shaped part of the plungers can pass variably through the recesses in the retainer bar [8] between the end of the syringe cylinder and the plunger endcaps until the springs (Alcomex D5470, Alcomex Federn GmbH, Goch, Germany) are brought in position. One side of the spring is equipped with a 3D-printed connector [6] that fits precisely on the endcap of the plunger, whereas the other side is equipped with a 3D-printed disk [2]. Attaching the spring to the endcaps of syringe plungers and placing the disks at the movement-transmission bar, the springs are under a preload of 6 N. If springs get compressed due to resistance of the plunger in the syringe cylinder, e.g., due to an agglomerate, the extent of compression can be directly measured by attached scales [5] or retrospectively evaluated from photos taken at defined time intervals by two webcams installed above the model and controlled via a Raspberry-Pi 2 computer. Syringes are held in position in the V-shaped recesses of the U-rail [12] by clamping the end of the cylinder [10] to the U-rail via an adapted rail.

#### 2.1.2. 3D Printed Parts 

The 3D printed parts were designed using FreeCAD 0.18. Stereolithography files were imported into the slicing software Cura (Version 3.6.0) and printed via an Ultimaker^3alpha extended^ 3D printer from FormFutura^®^ 3D Easyfill polylactic acid Printer Filament, all by Ultimaker (Utrecht, The Netherlands).

#### 2.1.3. Volume Pattern in Syringe Cavities

Volume changes in the cavities of the syringes where suspensions were once daily administered to, having been generated by movement of the plunger in the syringe cylinder, so that medium was drawn up and emptied together with non-agglomerated SD-ASD to the reservoirs directly attached to the syringes. To repeatably execute volume changes, we programmed a 23.5 h volume pattern using SyringePumpProV1 (Version 1.6.4.7). The frequency and dimension of motions and rests in this pattern shown in Figure 4 are freely chosen, the total amount of medium uptake of 37.5 mL, its distribution between the simulated inactive photophase (first twelve hours) and the active scotophase (last twelve hours) as well as uptake and emptying rates were transferred from physiological conditions.

The starting volume of the 23.5 h pattern was set at 3.5 mL and the end volume at 0.5 mL. Contrary to the constant medium uptake rate of 1.5 mL/min, we decreased the rate of medium emptying stepwise. Whereas 3% of the initial volume at the beginning of the emptying cycle were emptied per minute for the first 15 min, the rate decreased to 1.5%/min from 15–30 min and to 0.5%/min until the aspired volume of this emptying step was reached. In our experiments, suspensions were drawn up once daily to the syringes, where 30 min per day were planned, before the next cycle of 23.5 h could be started. Media in the reservoirs, deionized water adjusted to pH 3 using 1 N HCl, was replaced following daily administration of suspensions.

### 2.2. Tested Spray-Dried Formulations

Two HPMC-AS (Shin-Etsu AQOAT^®^ AS-LG)-based SD-ASDs were provided for in vitro testing by Boehringer Ingelheim Pharma GmbH & Co. KG, Biberach, Germany. Formulation 1 contained a drug load of 70% (*w/w*) BI 1026706 and formulation 2 of 80% (*w/w*) BI 1026891. Physicochemical properties of both crystalline NCEs and HPMC-AS-based SD-ASD formulations are given in Table 1. Bulk density was determined by carefully filling approximately 3 g of each ASD formulation in a 25 mL measuring cylinder. The particle density was evaluated with a AccuPyc II 1340 helium gas pyncometer (Micromeritics Instrument Corporation, Norcross, GA, USA). Particle size was measured using a Mastersizer 2000 equipped with a Hydro 2000 µP wet dispersion unit (Malvern Panalytical Herrenberg, Herrenberg, Germany), applying a stirrer speed of 1500 rpm and an ultrasound of 30% with water as dispersant.

Whereas formulation 1 showed high in vivo pharmacobezoar formation potential in studies with durations of daily dosing between 5 days and 26 weeks [6], firm pharmacobezoars were not found during necropsy of rats treated at nominally similar dose levels of formulation 2 for similar durations. However, findings of gritty material consisting of SD-ASD retained in the stomach at necropsy made this formulation interesting for investigation.

### 2.3. Preparation of Suspensions for In Vitro Testing 

Suspensions with a concentration of 100 mg/mL SD-ASD were prepared similar to the suspensions for preclinical testing in rodents. The appropriate amount of SD-ASD was weighed into a glass beaker, wetted by addition of a small amount of vehicle (0.01 N hydrochloric acid—HCl, pH 2), and manually stirred using a glass stick until a homogenous paste was formed. The remaining amount of vehicle was added and suspensions were further stirred at 200 rpm for 5 min using a magnetic stirrer (Heidolph MR 3001) and under continuous stirring drawn up into appropriate syringes. Vehicle was prepared every 4th day with 1% (*v/v*) 1 N hydrochloric acid in deionized water. In case of viscosity-enhanced vehicles, hydroxyethyl cellulose (Natrosol^®^ 250HX, Ashland, Wilmington, DC, USA) was added to the vehicle under rapid magnetic stirring prior to manufacturing the ASD suspension. Both the intermediate viscous vehicle containing HEC 0.5% (*w/w*) and the high viscous vehicle containing HEC 1% (*w/w*) were allowed to swell for 24 h in the fridge at 7 °C until both viscosity-enhanced vehicles were clear and homogenous. 

### 2.4. Conduction of In Vitro Experiments

Agglomeration potentials of suspensions were investigated by once daily administration of 2 mL of the suspensions containing the solid dispersion to be tested. This procedure is further referred to as “dosing” of the syringes. The suspensions were taken from the beakers, in which they were constantly stirred, before the syringes were placed in the model (Figure 5—step 1). Therefore, the upper part of the retainer bar was removed and the lower half of the cross-shaped section of the syringe plungers placed in the appropriate recesses. When the top of the bar was repositioned, the recesses enclosed the plungers circumferentially. Before media reservoirs were attached directly to the syringes and filled with 40 mL medium, syringe cylinders were firmly clamped to the backside of the U-rail, and a cover bar (not shown in Figure 2) was positioned above all syringes to fix the cylinders in their position in the V-shaped cutouts of the U-rail.

As the complex of the retainer bar and movement transmission bar was still in the final position of the last pattern (0.5 mL volume in syringes), it had to be brought back into the starting position (volume of 3.5 mL per syringe). Therefore, the pump executed a 3 mL draw up program. The plungers of syringes that were already dosed with 2 mL suspension slipped through the retainer bar until the endcaps were adjacent to the backside of the retainer bar. From there on, the plungers had to follow the movement of the retainer bar, thereby drawing up 1.5 mL medium from the reservoirs to the syringes (2 mL → 3.5 mL). Once back in starting position, springs had to be attached on the plunger’ endcaps and the volume pattern program was uploaded to the syringe pumps’ mainboard before the next 24 h cycle could be started. The experiment was terminated for those syringes in which an agglomerate in the syringe cylinder led to spring compression of at least 2 mm at position of minimal volume after running through the volume pattern. 

#### In Vitro Agglomerates

At the end of the pump cycle in which spring compression of at least 2 mm was observed, generated agglomerates were removed from the syringe. Therefore, syringe cylinders were cut open circumferentially with a scalpel as indicated in Figure 5, step 3. Agglomerates were placed on microscope glass slides and dried at 30 °C in a drying oven (Heraeus T20-Kendro, Langenfeld, Germany) until the 24 h loss-on-drying was lower than 1% (Figure 6 (below)).

The mass of the agglomerates per day as main comparative criterion was calculated by dividing the total mass of the dried in vitro agglomerate by the number of dosing days. As experiments were conducted in triplicate, results are given as the mean of agglomerated masses of formulation per day from three units ±  standard deviation. Statistical analysis was done applying an unpaired *t*-test using GraphPad Prism 5.01 and considering a significance level of *p* < 0.05.

## 3. Results and Discussion

### 3.1. In Vitro Model Development

The first step of the model development was to identify a cavity variable in volume, that would allow the simulation of a dynamic environment with adjustable fluid uptake and emptying pattern. Thus, three major reasons favored the application of a less complex approach regarding anatomical and morphological features compared to established in vitro models [16,17,18]. 

First, the comparative assessment of agglomeration potential prior to in vivo studies requires a high degree of repeatability, which should be achieved by reduction of complexity of the cavities’ spatial shape in which the agglomeration potential of solid dispersions is studied. 

Second, preliminary testing revealed that removal of agglomerates from cavities for further characterization would be either destructive to the compartment itself or likely cause unfavorable damage to the agglomerate formed. 

Third, valid statements about differences of SD-ASD agglomeration potential ideally require performing multiple experiments in parallel at an overall reasonable timescale to compare the agglomeration potential of different suspensions as a part of preclinical formulation development. Complex manufacturing of model components or complex preparation steps, as described for other rat stomach models, are therefore unsuitable and may furthermore transfer variability of model components, such as their volume [16,17,18], to the obtained results.

Compact cylindrical shape, variability in volume, and commercial availability in standardized quality plus an outlet diameter of 2 mm, that is comparable to the maximum size of particles which are able to pass the rat’s pyloric sphincter physiologically [19], led to the decision to apply 10 mL syringes with a central outlet as main compartment of the in vitro model.

The rationale for the design of the volume pattern was to transpose in vivo intragastric volume changes to media draw up and emptying processes in the model. In total, 37.5 mL medium was drawn up to each syringe per day. The first half of the volume pattern simulated the photophase, where activity of nocturnal rodents is low and thus frequency and amplitude of fluid uptake reduced. Fluid uptake of rodents in this period accounts for 20% of the daily intake, whereas in the active scotophase, simulated in the second half of the profile, the residual 80% is consumed (Figure 7). 

As mentioned, the amplitude and sequence of medium uptake and emptying steps were free chosen, however, with the total volume drawn up and the distribution to the simulated day (photophase) and night phase (scotophase) corresponding to physiological parameters [23]. Besides the volumes, the kinetics of filling and emptying of the syringes were adapted to physiologic rates of rats as well. In vivo fluid uptake proceeds with periods of activity and rest, controlled by a variety of factors [24,25]. However, the kinetics of drinking, consisting of multiple tongue laps per second, is known to be about 0.03 mL/s. As rests of few seconds were not considered for the filling process, we chose to apply a constant filling rate of 0.025 mL/s (1.5 mL/min) slightly lower than the reported in vivo rate. [23] As all plunger endcaps were in direct contact to the retainer bar when it was moved backwards, volumes of medium drawn up from the reservoir were fixed and equal in every syringe cavity.

Contrary to the burst and rest process of in vivo fluid uptake, gastric emptying of fluids from the stomach is a continuous process with a decreasing rate. To simulate the in vivo fluid emptying kinetics in vitro, we derived the percentage of fluid remaining in the stomach over time from in vivo profiles of fluid emptying from the rat stomach [16,20,21,22]. Based on these profiles, the percentage amount of medium emptied every 15 min following fluid uptake was calculated (Figure 7 (right)) and subsequently led to a series of 4, within these 15 min constant percentage rates. The percental rates (0–15 min; 15–30 min; 30–45 min; 45–60 min) were transferred to rates in mL/min based on the volume at the start of the emptying movement. From 30–45 min and 45–60 min, rates differed only marginally. The sequence of the 4 rates was run through until the volume specified in the volume pattern was reached. In cases where medium emptying exceeded 60 min, the last rate was applied until the intended volume was reached. This was restricted to emptying steps from the maximum volume of 3.5 mL to the minimal volume of 0.5 mL, taking 62 min. Depending on the agglomeration potential of formulations, a certain share of the administered dose was emptied with every emptying cycle. Even though these SD-ASD particles could theoretically be withdrawn again from the reservoir after they were emptied, we expect this amount to be negligible due to dilution and sedimentation in the media reservoirs, especially as the medium has been replaced every 24 h following daily administration of suspensions. 

The minimal volume of 0.5 mL, reached at several points of the volume pattern, was introduced to prevent complete emptying of syringes and to allow agglomeration of SD-ASD particles. If an agglomerate in the syringe cavity prevented a plunger from reaching the position of 0.5 mL, the respective spring, which transferred the forward movement of the movement transmission bar to the plunger endcaps, compensated for the resistance in the syringe cavity by compression (Figure 8). Spring compression of more than 2 mm after finishing the daily cycle signalized the completion of agglomerate formation. These syringes were removed from the model prior to the next dosing. If an agglomerate in one of the 18 syringes exceeded the minimal volume, the indirect movement transmission via springs ensured that this did not affect the other 17 syringes from following the volume pattern. Spring compression could be both visually monitored and retrospectively evaluated from photos taken by the cameras on top of the model in combination with the attached scales.

Deionized water adjusted to pH 3 with 1 N hydrochloric acid was used as medium in the reservoirs. This pH was chosen to simulate the acidic intragastric fluids as environment where in vivo pharmacobezoar formation takes place in, even though at the lower intragastric pH range. With respect to composition, a more sophisticated medium simulating the physiologically relevant fed state of rodents would be desirable [26,27].

In preclinical rodent studies, dose and administered volume are related to body weight of the animals. In vitro, we adopted this approach and standardized the dose of SD-ASD and the corresponding volume of suspension to simulate a rat with a body weight of 200 g. As standard dose, 1000 mg/kg/day SD-ASD suspended in 10 mL/kg vehicle, respectively 2 mL of a suspension containing 100 mg/mL SD-ASD in 0.01 N HCl was used as dose volume per syringe. Whereas the dosing volume meets recommendations for good practice in animal science [28,29], the dose was chosen due to its practical relevance for preclinical toxicological testing. A dose of 1000 mg/kg/day marks the maximum required dose to be tested if resulting plasma levels exceed 10-fold the expected clinical exposure when lower doses did not result in a mean exposure margin of 50-fold to the clinical exposure [1]. However, doses administered in vitro referred to mass of solid dispersions and did not consider drug load deviations of formulations.

### 3.2. Experiments on In Vitro Agglomeration Potential of Formulation 1 and Formulation 2 

In order to prove the capability of the in vitro model to detect differences of the agglomeration potential of solid dispersions, we tested suspensions of formulation 1 and formulation 2 in a sequence of three repetitive experiments with three units per formulation per experiment. 

Formulation 1 completed the formation of agglomerates within three days and obtained an agglomerated dry mass of 149.6 ± 19.2 mg/day (n = 9). Intraexperimental standard deviation of the three simultaneously investigated syringes was found between 1.2% to 5.3%. Spring compression indicated completion of agglomerate formation for formulation 2 after 4 days, with a mean agglomerated dry mass of 115.9 ± 11.3 mg/day (n = 9) and intraexperimental standard deviations of 5.6% to 7.8% (Figure 9). Interexperimental standard deviation of mean values of agglomerated mass per day from the three experiments was 12.8% for formulation 1 and 9.7% for formulation 2. The agglomeration potential of both formulations, indicated by the mean agglomerated mass per day, was statistically significantly different (*p* < 0.0006) (Figure 9—first and fourth column). 

In addition, we investigated the impact of the viscosity of the vehicle on the agglomeration potential of both formulations (each n = 3). Using the intermediate viscous vehicle (0.5% HEC in 0.01 N HCl) to suspend SD-ASD formulations, the in vitro agglomeration potential of formulation 1 was found to be reduced by 36%, and 24% for formulation 2. The effect was even more pronounced using the high viscous vehicle (1% HEC in 0.01 N HCl), resulting in reduction of agglomerated mass per day of 88% for formulation 1 and 87% for formulation 2. In case of the latter, formation of agglomerates inducing spring compression took up to 14 days.

Differences of agglomerated masses per day between both formulations were especially pronounced without viscosity enhancer added to the vehicle and are in line with the in vivo observed higher pharmacobezoar formation potential of formulation 1 compared to formulation 2 [6]. This difference is even more remarkable considering the similar physicochemical properties of both formulations (Table 1). While it would be conclusive that, e.g., differences in density of the vehicle and solid dispersion accelerate phase separation and therefore favor pharmacobezoar formation, none of the physicochemical properties theoretically indicates the predisposition of formulation 1. However, to state a correlation between physicochemical properties and agglomeration potential with certainty, further in vitro testing of more solid dispersions beyond the conducted experiments is necessary to identify factors that significantly influence the agglomeration potential of solid dispersions as well as the extent of correlation to the pharmacobezoar formation potential in vivo. 

In terms of transferability of in vitro results to in vivo, the fact that agglomerates were obtained with formulation 2 as well shows that the processes of pharmacobezoar formation and in vitro agglomeration of SD-ASDs might not be exactly transferable to each other. In vivo, pharmacobezoar formation taking place in a continuously changing environment in terms of shape, motility, pH, and presence of gastric contents until end of dosing has been highly variable from one nonclinical study to the other [6]. To obtain meaningful results, we transferred this process to a dynamic fluid environment with agglomerate formation is the intended endpoint, and thereby reduced variability of the agglomeration of solid dispersions. Generally, the in vitro model should be understood as a tool to compare the agglomeration potential of suspensions of solid dispersions in order to estimate the probability of pharmacobezoar formation and not as a tool to simulate the process of pharmacobezoar formation in vivo.

Referring to the physical background of agglomeration processes, it is known that enhanced particle motion and accompanying collisions enhance aggregation of particles [30]. Through the developed model able to study the effect of hindered particle motion due to viscosity enhancement of the vehicle, the significance of the effect was with a reduction of agglomerated mass by more than 85% impressive. Although 0.5% HEC might be the more feasible concentration, 1% HEC is reported to be practical as well to increase stability of suspensions in terms of homogeneity and dose uniformity of suspensions for oral administration to rodents [29,31].

## 4. Conclusions

Unawareness of the pharmacobezoar formation potential of SD-ASD-containing suspensions in the stomach of rodents in nonclinical safety studies includes the risk of adverse events affecting animal welfare and thus, the potential to misinterpret experimental results. Addressing this problem, we developed an in vitro model enabling parallel comparative assessment of the agglomeration potential of up to 6 different suspensions, each in triplicate.

Based on the study design of preclinical oral toxicity studies, in vivo pharmacobezoar formation in the rat stomach following once daily oral administration of suspensions has been transposed to once daily administration of suspensions to syringes, whose volume in the cylinder is changed by uptake and emptying of medium. Draw up and emptying processes were adapted to physiological volumes and rates. To allow formation of agglomerates in the syringe cylinder, a minimum volume of 0.5 mL is mandatory. Concentration and volume of SD-ASD suspensions administered daily to the syringe cavities were adapted from standards of preclinical practice and related to a hypothetical body weight of 200 g per rat. When agglomerates in the syringes exceeded the minimal volume and thus induced spring compression, syringe cylinders were cut open to extract the agglomerates. Their dry mass divided by the number of daily dosings of SD-ASD suspensions equaled the agglomerated mass per day as outcome criterium.

To prove the in vitro model’s ability to detect differences in agglomeration potential, a sequence of three experiments on two HPMC-AS-based SD-ASD formulations has been conducted to investigate their agglomeration potential and its intra- and interexperimental variability. We found that the in vitro agglomeration potential of formulation 1 and formulation 2 correlated with their in vivo pharmacobezoar formation potential observed in previous preclinical testing. Subsequently, we tested the influence of the vehicles’ viscosity, used to suspend the SD-ASDs, on in vitro agglomeration of both formulations. We observed a significant decrease of agglomerated mass/day with increasing concentration of viscosity enhancer in the vehicle used to prepare suspensions for both formulations. Thus, increasing the vehicles’ viscosity might be an approach to reduce the pharmacobezoar formation potential of SD-ASDs in vivo.

The developed in vitro model should be considered as a tool to assess the pharmacobezoar formation potential of suspensions of solid dispersions based on their in vitro agglomeration potential so that action can be taken to prevent pharmacobezoar-induced adverse effects on animal welfare.

## Figures and Tables

**Figure 1 pharmaceutics-14-02608-f001:**
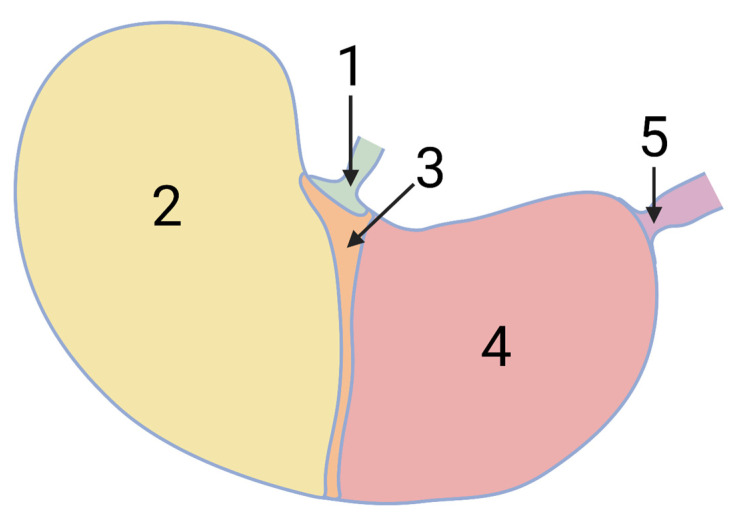
Schematic illustration of the rat stomach’s anatomy (1—cardia; 2—nonglandular stomach; 3—limiting ridge; 4—glandular stomach; 5—pylorus).

**Figure 2 pharmaceutics-14-02608-f002:**
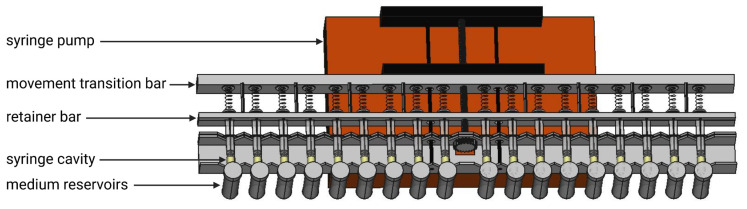
Schematic top view on the in vitro model.

**Figure 3 pharmaceutics-14-02608-f003:**
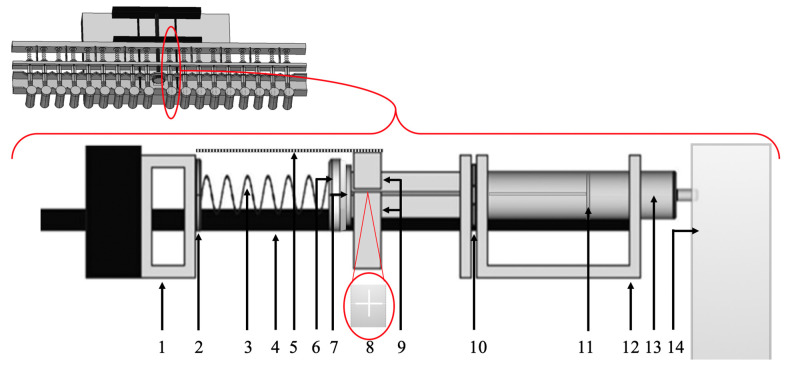
Schematic top view on the in vitro model (top). Cross section of the in vitro model with one unit displayed (below). (1—movement transmission bar; 2—disk; 3—spring; 4—spindle; 5—scale; 6—connector; 7—plunger endcaps; 8—recesses for syringe plungers; 9—retainer bar; 10—holder bars; 11—syringe plunger; 12—U-rail; 13—cavity of the syringe cylinder; 14—media reservoir).

**Figure 4 pharmaceutics-14-02608-f004:**
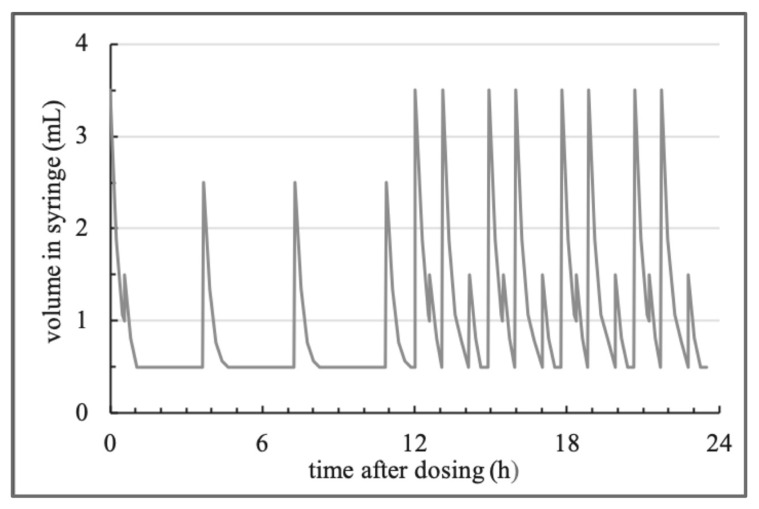
Volume in the syringe cavities over time.

**Figure 5 pharmaceutics-14-02608-f005:**
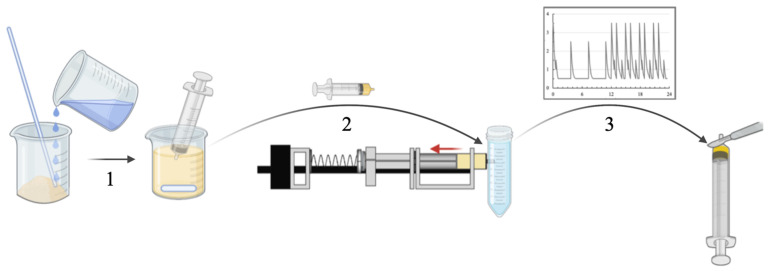
Graphical illustration of dosing procedure 1—transferring suspensions to syringe; 2—placing syringe in the in vitro model; 3—repeating daily dosing procedure (steps 1 and 2) including completion of the volume pattern until agglomerate formation is completed and agglomerates can be extracted by cutting the syringe cylinder with a scalpel.

**Figure 6 pharmaceutics-14-02608-f006:**
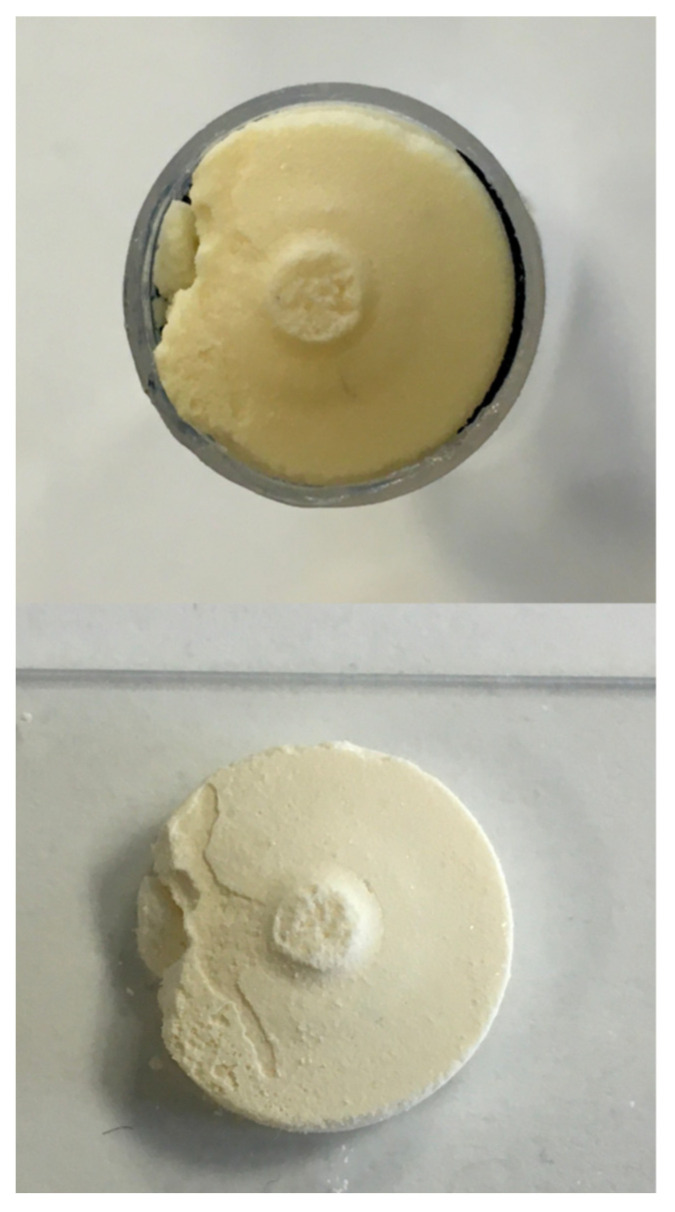
Agglomerate of formulation 1 in the opened syringe cylinder (top) and following drying on a microscopic slide (below).

**Figure 7 pharmaceutics-14-02608-f007:**
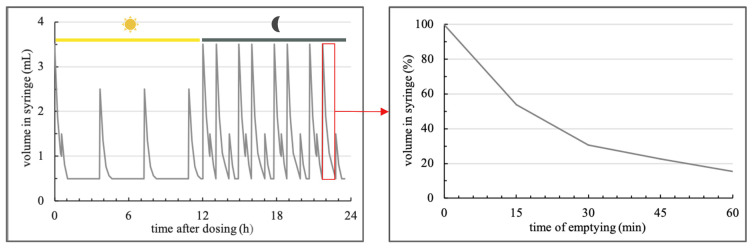
Volume pattern with marked photo and scotophase (**left**) and one hour kinetics of emptying derived from in vivo data [16,20,21,22] (**right**).

**Figure 8 pharmaceutics-14-02608-f008:**
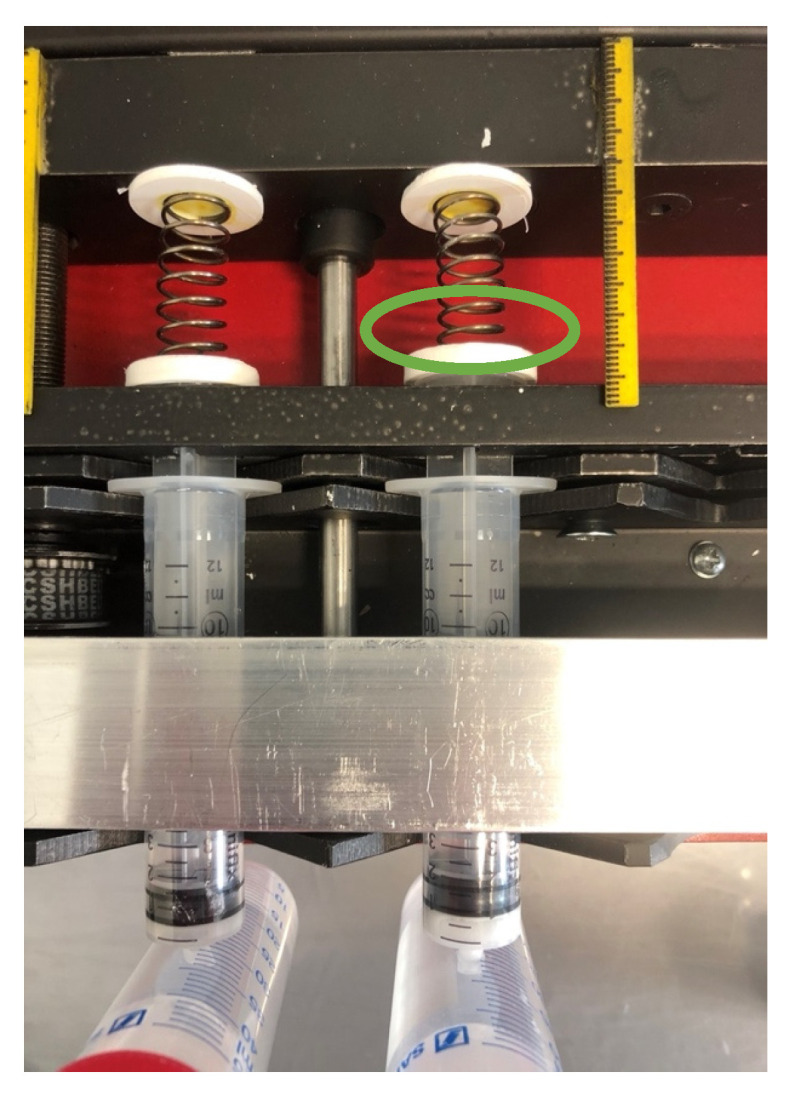
Top view on two units in the in vitro model—completion of agglomerate formation in the right syringes is indicated by the compression of the spring (indicated by green cycle).

**Figure 9 pharmaceutics-14-02608-f009:**
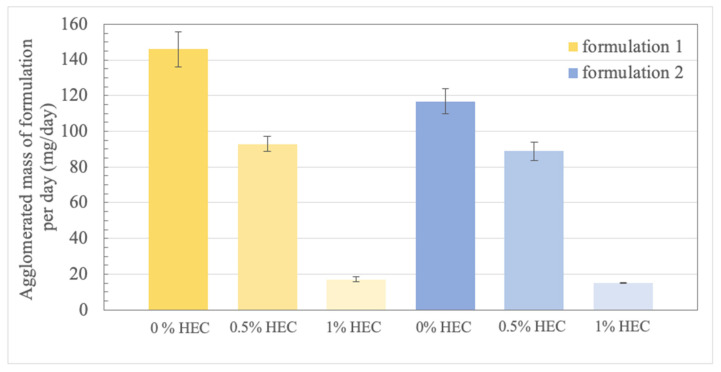
In vitro agglomerated masses per day of formulation 1 (yellow) and formulation 2 (blue) from suspensions without (n = 9), with 0.5% and 1% (*w/w*) hydroxyethyl cellulose (HEC) added to the vehicle as viscosity-enhancer (both n = 3).

**Table 1 pharmaceutics-14-02608-t001:** Physicochemical parameters of crystalline NCEs and corresponding spray-dried formulations applied in preclinical testing.

NCE	BI 1026706	BI 1026891
pKa (Acid/Base)	8.6 (Acid)	4.4 (Base)
Water solubility pH 7.4 (mg/mL)	0.02	0.002
Intrinsic dissolution rate (µg/cm^2^*min)	<50	<20
LogP	3.2	3.8
**Solid dispersion**	**Formulation 1**	**Formulation 2**
Drugload (%)	70	80
Spray-drying solvent	Methanol	Acetone/H_2_O 80/20 (%, m/m)
Bulk density (g/cm^3^)	0.33	0.37
Particle density (g/cm^3^)	1.45	1.51
D50 (µm)	6.61	5.17

## Data Availability

Not applicable.

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
