# Peer review of "An In Vitro Model to Investigate the Potential of Solid Dispersions to Form Pharmacobezoars"

_pharmaceutics, 2022, doi:10.3390/pharmaceutics14122608_

Round 1
Reviewer 1 Report
The manuscript entitled "An in vitro model to investigate the potential of solid dispersions to form pharmacobezoars" is a well-written and well-researched article. The authors have clearly described their in vitro model and how it can be used to study the potential of solid dispersions to form pharmacobezoars. The results of the study are clearly presented and discussed. However, there are a few areas that could be improved.
First, the authors should provide more information on the potential clinical relevance of their findings. It is not clear why pharmacobezoars are a problem that needs to be addressed or how this in vitro model can help to solve this problem.
Second, the authors should provide more details on the methodology in terms of reference used. For example, the authors should provide suitable reference for the section 2.3 Preparation of suspensions for in vitro testing and section, 2.4 Conduction of in vitro experiments
Third, the authors should discuss the limitations of their study. For example, the authors should acknowledge that in vitro model may not completely reflect the in vivo situation.
Overall, this is a well-written and well-researched article. However, the authors should provide more information on the clinical relevance of their findings.
Author Response
Answers to comments and suggestions of reviewer 1
Dear reviewer, we appreciate your efforts in order to improve our manuscript and the valuable comments.
First, the authors should provide more information on the potential clinical relevance of their findings. It is not clear why pharmacobezoars are a problem that needs to be addressed or how this in vitro model can help to solve this problem.
We added on page 2 line 57-61 the symptomatic induced by pharmacobezoars in rats (“…but induced besides histopathological findings in the stomach no relevant symptomatic. However, acute symptomatic was observed when pharmacobezoars emptied from the stomach induced an obstructive ileus, which required unscheduled euthanasia of three female rats in a 13—week oral toxicity study. ….”). For detailed information on incidence and symptomatic please see the manuscript “Pharmacobezoar Formation from HPMC‑AS‑containing Spray‑dried Formulations in Nonclinical Safety Studies in Rats” which is cited in this manuscript as it has been accepted recently by the journal Toxicologic Pathology and will be published soon. We added this manuscript in the attachment as separate pdf document. As the observed acute symptomatic induced by pharmacobezoars showed that there is a need to investigate the pharmacobezoar formation potential of SD-ASD prior nonclinical studies in rodents, we developed the in vitro model introduced in this work as stated on page 2 line 85 - page 3 line 97
Second, the authors should provide more details on the methodology in terms of reference used. For example, the authors should provide suitable reference for the section 2.3 Preparation of suspensions for in vitro testing and section, 2.4 Conduction of in vitro experiments
We searched to describe methodology step by step in a way that enables the reader to reconduct the presented experiments. Whereas suspensions of both tested SD-ASD formulations were prepared similar to the way of preparing suspensions for nonclinical testing in rodents (section 2.3), the procedure of conducting of in vitro experiments has been developed parallel to the in vitro model development itself and was conducted as described in section 2.4 in all in vitro experiments. To clarify the link between the preparation of suspensions in toxicological studies in vivo and the preparation of suspensions for in vitro testing, we added on page 6 line 242- page 7 line 250: “… SD-ASD were prepared similar as the suspensions for preclinical testing in rodents. The appropriate amount of SD-ASD has been weighed into a glass beaker, wetted by addition of a small amount of vehicle (0.01 N hydrochloric acid – HCl, pH 2) and manually stirred …”.
Third, the authors should discuss the limitations of their study. For example, the authors should acknowledge that in vitro model may not completely reflect the in vivo situation.
We included a section in the discussion part of the manuscript on page 12 from line 462-473 where we searched to explain the differences between in vitro agglomeration and pharmacobezoar formation. We searched to clarify the addressed point by closing the discussion at page 12 line 470-473 with “Generally, the in vitro model should be understood as a tool to compare the agglomeration potential of suspensions of solid dispersions in order to estimate the probability of pharmacobezoar formation and not as a tool to simulate the process of pharmacobezoar formation in vivo.” and repeated this in the last statement of the conclusion.
Overall, this is a well-written and well-researched article. However, the authors should provide more information on the clinical relevance of their findings.
We agree that it is somewhat difficult to see the clinical relevance without having the detailed information on the findings given in our accepted manuscript “Pharmacobezoar Formation from HPMC‑AS‑containing Spray‑dried Formulations in Nonclinical Safety Studies in Rats” (to be published in Toxicologic Pathology).

Reviewer 2 Report
Regarding the manuscript (pharmaceutics- 2036213) entitled:
“An in vitro model to investigate the potential of solid dispersions to form pharmacobezoars”
General comment
The study presented a way to evaluate the risk of pharmacobezoar formation before in vivo administration. This study tackled a unique point. The manuscript, in general, is well written with sufficient data that proved the aim of the study.
Comments:
- Can the author correlate the generated data with an in-vivo study to validate the developed in vitro model?
- To validate this model different pharmacobezoar should be tested.
Author Response
Answers to comments and suggestions of Reviewer 2
Dear reviewer, we appreciate your efforts in order to improve the manuscript and the valuable comments.
Can the author correlate the generated data with an in-vivo study to validate the developed in vitro model?
To validate this model different pharmacobezoar should be tested.
Referring to both comments, it is important to note that formulation 1 is to the best of our knowledge the only SD-ASDs that formed pharmacobezoars during nonclinical testing in rodents at an incidence that allows drawbacks on the time and dose dependency of pharmacobezoar formation. Thus, it is the only SD-ASD that can be referred to for any correlations to in vivo pharmacobezoar formation. For detailed information on incidence and induced symptomatic please see the manuscript “Pharmacobezoar Formation from HPMC‑AS‑containing Spray‑dried Formulations in Nonclinical Safety Studies in Rats” which is cited in this manuscript as source 6. The manuscript has been accepted recently by the journal Toxicologic Pathology and will be published soon. We added this manuscript as a separate pdf document in the attachment.
In the manuscript reviewed now, we searched to validate the ability of the in vitro model by testing two formulations with known pharmacobezoar formation potential. As mentioned, we found a statistically relevant reduction of in vitro agglomerated mass of formulation 2 per day compared to formulation 1 (Figure 9 and page 11 line 431 – page 12 line 440), that is reflected by the absence of pharmacobezoar formation in the preclinical testing program of formulation 2. Thus, the in vitro model proofed to be suitable to detect relevant differences of the agglomeration potential between both these formulations. We agree that further SD-ASDs for in vitro testing would be helpful to determine the extent of correlation between the agglomerated mass per day and the pharmacobezoar formation potential in vivo. We added this aspect to the statement on Page 12 line 448-461 (“However, to state a correlation between physicochemical properties and agglomeration potential with certainty, further in vitro testing of more solid dispersions beyond the conducted experiments is necessary to identify factors that significantly influence the agglomeration potential of solid dispersions and the extent of correlation to the pharmacobezoar formation potential in vivo.”).
By gaining attention for the problem of pharmacobezoar formation, we hope to get access to other formulations than the introduced formulation 1, that also showed a certain extent of pharmacobezoar formation in vivo to extend the range of formulations that can be used to validate the in vitro model. We would also like to point out that the in vitro model has been used to assess the influence of the viscosity of the vehicle used to suspend SD-ASDs in and that the observed reduction of the agglomeration potential will be validated in vivo in a follow up study as indicated on page 13 line 513-518.

Reviewer 3 Report
Dear authors,
thanks for this very interesting paper I have mainly two topics which I would like to address with my comments. Both are linked to the in vivo data.
1) Introduction: Line 52ff. "Besides few cases in mice, rats were particularly affected of pharmacobezoar formation following repeated oral dosing of suspensions of HPMC-AS based ASDs." It would be extremely helpful for the reader if you provide some literature references.
2) In the results and conclusion section you provide little insight into the in vivo data (you mention Gierke et. al 2022 as reference, but it does not appear in the list of references yet. It is therefore very difficult to judge/understand how good your in vitro assessment reflects the in vivo situation.
In addition, you could provide more clarity on the lengths of the test. In line 267ff you only mention when you stop the investigation (in case agglomerates are formed), but you don't mention how long you would test in case no agglomerates are formed (e.g. would you test for the same duration as the tox study? This could be quite long and might not be reasonable as a pretest.).
Furthermore, could the difference in agglomerated mass of formulation per day (Figure 9) be related to the drug load, e.g. lower drug load, higher amount of polymer.
Despite these comments I realized the different notation for the reference Gierke et. al 2022 in comparison to all other references. Please check the manuscript accordingly. In a few cases cross-references to figures/tables didn't work, e.g. line 185, 208, 423. Typos are present in line 195 (initial volume) and 328 (is known). Please correct.
Author Response
Answers to comments and suggestions of Reviewer 3
Dear reviewer, we appreciate your efforts in order to improve the manuscript and the valuable comments.
1) Introduction: Line 52ff. "Besides few cases in mice, rats were particularly affected of pharmacobezoar formation following repeated oral dosing of suspensions of HPMC-AS based ASDs." It would be extremely helpful for the reader if you provide some literature references.
We added on page 2 line 49 “ … to the previously reported formation …” so that in the introduction is now directly referred to our previous paper that is to the best of our knowledge the only report on pharmacobezoars in the context of preclinical testing in rodents. This manuscript has been accepted recently by the journal Toxicologic Pathology and will be published soon. To enable access to details as long as we wait for publication, we added this manuscript as separate pdf document in the attachment.
2) In the results and conclusion section you provide little insight into the in vivo data (you mention Gierke et. al 2022 as reference, but it does not appear in the list of references yet. It is therefore very difficult to judge/understand how good your in vitro assessment reflects the in vivo situation.
We adapted notation of the mentioned paper to the citation style in the reviewed manuscript. Furthermore, we searched to clarify the differences between in vitro agglomeration and pharmacobezoar formation in vivo on page 12 line 462-473 (“Generally, the in vitro model should be understood as a tool to compare the agglomeration potential of suspensions of solid dispersions in order to estimate the probability of pharmacobezoar formation and not as a tool to simulate the process of pharmacobezoar formation in vivo.”). This message is also repeated at the end of the conclusion.
In addition, you could provide more clarity on the lengths of the test. In line 267ff you only mention when you stop the investigation (in case agglomerates are formed), but you don't mention how long you would test in case no agglomerates are formed (e.g. would you test for the same duration as the tox study? This could be quite long and might not be reasonable as a pretest.).
Please see page 11 line 413 as well as page 12 line 416 and 430, where the experiment duration until agglomerates exceeded the minimal volume are given.
Furthermore, could the difference in agglomerated mass of formulation per day (Figure 9) be related to the drug load, e.g. lower drug load, higher amount of polymer.
To certainly answer this interesting question, we would need to test more SD‑ASDs as mentioned at page 12 line 449-461. However, with only one formulation (formulation 1) with known high pharmacobezoar formation potential in vivo, there is a lack of SD-ASD formulations to evaluate the reasons why some SD-ASDs obtain extensive agglomeration, respectively pharmacobezoar formation potential. By raising attention for the problem of pharmacobezoar formation, wehope to get access to other formulations that showed a certain extent of pharmacobezoar formation in vivo. This work should first off introduce the in vitro model, show its capability of detecting differences in the agglomeration potential of two SD-ASD formulations and present an approach to reduce the agglomeration potential of SD-ASDs. The correlation of physicochemical parameters of SD-ASDs to their agglomeration potential would be an interesting follow up study the introduced model can be applied for.
Despite these comments I realized the different notation for the reference Gierke et. al 2022 in comparison to all other references. Please check the manuscript accordingly. In a few cases cross-references to figures/tables didn't work, e.g. line 185, 208, 423. Typos are present in line 195 (initial volume) and 328 (is known). Please correct.
Thank you, recommended corrections are made.

Round 2
Reviewer 2 Report
Accepted